# Genome-Wide Identification and Characterization of Xyloglucan Endotransglycosylase/Hydrolase in *Ananas comosus* during Development

**DOI:** 10.3390/genes10070537

**Published:** 2019-07-16

**Authors:** Qingyun Li, Huayang Li, Chongyang Yin, Xiaotong Wang, Qing Jiang, Rui Zhang, Fangfang Ge, Yudong Chen, Long Yang

**Affiliations:** 1Agricultural Big-Data Research Center and College of Plant Protection, Shandong Agricultural University, Taian 271018, China; 2Center for Genomics and Biotechnology, Haixia Institute of Science and Technology, Fujian Provincial Key Laboratory of Haixia Applied Plant Systems Biology, Fujian Agriculture and Forestry University, Fuzhou 350002, China

**Keywords:** xyloglucan endotransglycosylase/hydrolase, *Ananas comosus*, F153, MD2, crassulacean acid metabolism

## Abstract

Xyloglucan endotransglycosylase/hydrolase (XTH) is a cell-wall-modifying enzyme participating in diverse cell morphogenetic processes and adaptation to stress. In this study, 48 XTH genes were identified from two pineapple (*Ananas comosus*) cultivars (‘F153’ and ‘MD2’) and designated *Ac(F153)XTH1* to *-24* and *Ac(MD2)XTH1* to *-24* based on their orthology with *Arabidopsis thaliana* genes. Endoglucanase family 16 members were identified in addition to XTHs of glycoside hydrolase family 16. Phylogenetic analysis clustered the XTHs into three major groups (Group I/II, III and Ancestral Group) and Group III was subdivided into Group IIIA and Group IIIB. Similar gene structure and motif number were observed within a group. Two highly conserved domains, glycosyl hydrolase family 16 (GH16-XET) and xyloglucan endotransglycosylase C-terminus (C-XET), were detected by multiple sequences alignment of all XTHs. Segmental replication were detected in the two cultivars, with only the paralogous pair *Ac(F153)XTH7-Ac(F153)XTH18* presented in ‘F153’ prior to genomic expansion. Transcriptomic analysis indicated that XTHs were involved in the regulation of fruit ripening and crassulacean acid metabolism with tissue specificity and quantitative real-time PCR analysis suggested that *Ac(MD2)XTH18* was involved in root growth. The results enhance our understanding of XTHs in the plant kingdom and provide a basis for further studies of functional diversity in *A. comosus*.

## 1. Introduction

Pineapple (*Ananas comosus*), a member of the Bromeliaceae family, is cultivated widely in tropical and subtropical regions and is renowned for its nutritional and medicinal values [1]. Given its status as a herbaceous perennial monocotyledon with classical crassulacean acid metabolism (CAM), the pineapple genome was sequenced in 2015 as a model plant [2]. The genome of the cultivar ‘MD2’, which is the predominant pineapple cultivar grown worldwide by virtue of its fruit flesh flavor and commercial value, and the cultivar ‘F153’ has been sequenced [3]. Pineapple genome sequencing has provided valuable information for further research for crop improvement [4,5].

Xyloglucan endotransglycosylase/hydrolase (XTH) participates in diverse physiological processes, especially cell elongation and stress resistance [6]. XTH is a cell-wall-modifying enzyme encoded by a multigene, which belongs to a subfamily of the glycoside hydrolase family 16 (GH16) [7,8]. Generally, XTH proteins perform two diverse catalytic activities: xyloglucan endohydrolase (XEH) and xyloglucan endotransglycosylase (XET) [9,10]. XET activity is characterized by the no-hydrolytic cleavage and rejoining of xyloglucan (XyG) chains, whereas XTH activity irreversibly cleaves hydrolytic XyG chains to promote the expansion, degradation, remediation, and morphogenesis of the cell wall [6,9]. To date, the majority of identified XTH proteins show XET activity [6]. XTH proteins share the conserved glycosyl hydrolase family 16 domain (GH16_XET) with a specific EXDXE motif likely to be the catalytic site for both XET and XEH activities. XTH proteins also contain the significant xyloglucan endotransglycosylase C-terminus domain (C-XET), which distinguishes XTH proteins from other GH16 subfamilies [6,11,12]. The XTH gene subfamily was originally divided into three major groups, of which Group III was subdivided into subgroups IIIA and IIIB [10,13]. With the expansion of XTH observations, more detailed clade and subclade groupings (Group I/II, IIIA, IIIB, and an Ancestral Group) were applied to different species on the basis of sequence similarity [10,11]. Interestingly, XTH genes predominantly display XET activity in Group I/ II and IIIB, whereas XEH activity is characteristic of Group IIIA [13]. In *Arabidopsis thaliana*, Group IIIA genes are ubiquitous and dispensable in plant growth [14].

An increasing number of XTHs have been identified using publicly available datasets [15]. For instance, 33, 29, 25, 56, 61, and 24 potential XTH members have been defined in *A. thaliana*, *Oryza sativa*, *Solanum lycopersicum*, *Nicotiana tabacum*, *Glycine max*, and *Hordeum vulgare*, respectively [5,16,17,18,19,20]. In *A. thaliana*, XTHs show distinct and diverse organ-specific expression patterns. Five genes (*AtXTH-1*, *-21*, *-22*, *-30*, and *-33*) were expressed preferentially in green siliques, whereas two genes (*AtXTH-24* and *-32*) were expressed primarily in stems [16]. In *O. sativa*, seven root-specific XTH genes (*OsXTH1*, *-2*, *-4, -13*, *-15*, *-16*, and *-25*) were predominantly expressed in roots of 14-d-old seedling, whereas no expression was detected in other tissues [17]. XTH proteins modify the complex structure of lignin and cellulose in a variety of developmental processes, such as root formation, flower generation, and fruit softening [5,21,22]. Numerous XTHs have been detected in root elongation zones and trichoblasts of diverse vascular plants [23]. Compared with other tissues in *Dianthus caryophyllus*, *DcXTH2* and *DcXTH3* transcripts were markedly accumulated in petals and showed XET activity during flower opening stages [22]. *SlXTH5* detected at multiple stages during tomato fruit’s expansion and displayed XET activity in concordance with results for apple, kiwifruit, and strawberry [24,25]. Thus, XTH proteins play a critical role during fruit growth and ripening. In addition, XTHs show abnormal expression under abiotic stress. The XTH gene *CaXTH3* of *Capsicum annuum* showed a high expression level in transgenic *A. thaliana* lines and conferred enhanced salt and drought tolerance [26]. *AtXTH14*, *-15*, and *-31* showed remarkably low expression under aluminum treatment in *A. thaliana* roots, especially *AtXTH31* [27]. In contrast, *MtXTH3* was strongly up-regulated by a higher NaCl concentration in *Medicago truncatula* [28]. Thus, XTHs possess considerable spatial and temporal specificity, and respond to a variety of environmental stimuli for adaptation cell wall enzyme activities.

Additional potential members of the XTH gene family can be identified by genome-wide analysis using the published genome resources in *silico*. Systematic identification and characterization of XTHs in pineapple have received only limited attention to date [15]. In this study, we conducted a comprehensive analysis, including the classification, evolutionary relationships, and expression patterns of XTHs to determine whether pineapple XTHs participate in the CAM pathway and perform important functions in the leaf and root. This research provides novel insights into the functional characteristics of XTHs in pineapple and will facilitate further study of the regulatory mechanism at different developmental stages.

## 2. Materials and Methods

### 2.1. Dataset Compilation and Identification of the XTH Gene Family 

The protein sequences of *A. comosus* (L.) Merry two cultivars: ‘F153’ and ‘MD2’ were downloaded from the National Center for Biotechnology Information database (accession numbers are GCA-001540865.1 and GCA-001661175.1, respectively) [29]. The protein sequences for *A. thaliana* were obtained from The Arabidopsis Information Resource [30]. The protein ID of XTHs in *A. thaliana* as reference sequences collected from a former publication [9]. 

Two Hidden Markov Models (HMMs) were downloaded from Pfam and used to scan XTH sequences using the default E-value in HMMER 3.0 [31,32]. The HMM profile established for *AtXTH* genes was used to search for candidate XTH family members in pineapple. Potential XTH protein sequences were further detected using BLASp. Candidate genes were filtered and identified using the Conserved Domain Search Service (CD-Search) [33]. The length, molecular weight (MW), and theoretical isoelectric point (PI) of XTHs were characterized with ExPASy [34]. Their single peptide and subcellular localization were predicted by SignalP 4.1 and Plant-mPLoc [35,36].

### 2.2. Multiple Sequence Alignment and Identification of Motifs

A multiple sequence alignment of the candidate pineapple XTH proteins was generated and obvious features of the sequences were displayed using ClustalX2 with the default options [37]. In addition, motifs were detected using Multiple Expectation Maximization for Motif Elicitation with a motif width of 6–50 residues and a maximum of 10 motifs [38].

### 2.3. Phylogenetic Tree Analysis and Nomenclature of XTHs 

Multiple sequence alignments of the XTH proteins from pineapple and *A. thaliana* were generated using the ClustalW with default parameters (pairwise alignment with gap opening penalty of 10 and gap extension penalty of 0.1, multiple alignment parameters with gap opening penalty of 10, a gap extension penalty of 0.2, and delay divergent sequences set at 30%) [37]. A phylogenetic tree was constructed using the neighbor-joining algorithm with 1000 bootstrap replications using MEGA7 [39]. All members were numbered sequentially and designated as *Ac(F153)XTH* or *Ac(MD2)XTH* based on the genotypic origin of the gene [40].

### 2.4. Gene Structure Analysis 

The gene structures of the candidate XTHs were predicted using the online software Gene Structure Display Server [41]. The complex figure including the phylogenetic tree, gene structure, and motif distribution was arranged correctly using TBtools [42].

### 2.5. Chromosomal Distribution and Gene Duplication

All *Ac(F153)XTH* genes were localized on chromosomes based on their physical coordinates using MapChart and Perl script [43]. The Multiple Collinearity Scan toolkit was employed to identify syntenic blocks in the two pineapple genome assemblies and Circos software was used to depict the collinearity relationships [44,45]. 

### 2.6. Calculation of Ka/Ks

The synonymous substitution (*Ka*) and non-synonymous substitution (*Ks*) of XTH pairs were calculated using Ka/Ks Calculator 2.0 by the Nei and Gojobori (NG) method [46]. Fisher’s exact test was applied to confirm the validity of the ratio. To estimate the selection pressure, a ratio of *Ka/Ks* greater than one, equal to one, and less than one displayed positive selection, neutral selection, and purity selection respectively. The divergence time (*T*) was calculated as *T* = *Ks*/(2 × 6.1 × 10^−9^) × 10^−6^ million years ago (Mya).

### 2.7. Transcriptome Analysis and Gene Expansion Patterns 

Pineapple transcriptome data were downloaded from the Pineapple Genomics Database consisting of data from fruit at five developmental stages, the green leaf tip, and white leaf base tissues. The stages Fruit1 to Fruit 5 were ordered chronologically and represented the entire period of fruit ripening. Leaves were harvested from plants of the cultivar ‘MD2’ at 13 time points over a 24-h period [47]. Daytime is from 6.a.m to 4.p.m. and nighttime is from 4.p.m. to 6.a.m. in this study. The white leaf base comprised non-photosynthetic tissues sensitive to sunlight and the green leaf tip represented photosynthetic tissues. The transcript levels were visualized using R software.

### 2.8. Experimental Validation of XTH Transcript Levels by RT-qPCR Analysis

The green leaf tip, white leaf base, and root of ‘MD2’ plants were sampled at the Fujian Agriculture and Forestry University. All samples were rapidly frozen in liquid nitrogen then stored at −80 °C. Total RNA was extracted from each sample using a RNA extraction kit (Roche Diagnostics GmbH, Mannheim, Germany), then stored at −80 °C until further analysis. Quantitative real time PCR (RT-qPCR) analysis was performed using the BIO-RAD CFX Connect^™^ Real-time PCR Detection System with three biological replicates per sample. The transcript levels were analyzed using the 2^−ΔΔCt^ method and means ± standard errors (SE). The primer sequences used are presented in Appendix A. 

## 3. Results

### 3.1. Identification and Characteristics of XTHs

Twenty-eight potential XTH family members were detected with the two HMM models of the GH16_XET domain with a β-jelly-roll topology and XET_C domain, and 24 proteins were detected by BLASTp in the ‘F153’ reference genome. Candidates were confirmed to contain two highly conserved domains using CD-Search. Redundant proteins were manually removed on account of the absence of characteristic amino acid residues in the C-terminal region or a lack of the conserved motif ExDxE [12]. Finally, 24 candidates were identified in ‘F153’ and were designated *Ac(F153)XTH1* to *-24* based on homology with the classification of *A. thaliana* (Figure 1). In the same manner, 24 proteins were identified in ‘MD2’ and designated *Ac(MD2)XTH1* to *-24* in ‘MD2’. In addition, EG16 homologs, which are related to XTH members of the GH16 family but lack the XET_C extension, were identified in cultivars ‘F153’ and ‘MD2’ [48].

The candidate XTHs exhibited similar properties, including length, molecular weight (MW), isoelectric point (PI), and signal peptide (SP) (Table 1). Comparison of the length of the 48 XTH proteins revealed that *Ac(MD2)XTH5* was the largest protein with 575 amino acids, and the smallest one was *Ac(MD2)XTH1* with 230 amino acids. The MW ranged from 25.99 kDa to 64.58 kDa and corresponded with the protein length. Owing to the complex amino acid polarity, the PI ranged from 4.68 to 9.53. Subcellular localization prediction revealed that each XTH was localized to the cell wall, and 14 (7/7, ‘F153’/‘MD2’) proteins were targeted in both the cell wall and cytoplasm. The majority of the proteins contained signal peptide sequences (Appendix A).

### 3.2. Phylogenetic Analysis and Classification of XTH Proteins

A phylogenetic tree representing the relationships among 81 (24/24/33) XTHs of ‘F153’, ‘MD2’ and *A. thaliana* was constructed. The XTHs were clustered into three main groups (Group I/II, III, and Ancestral Group) (Figure 1). The majority of groups comprised the same number of XTHs in ‘F153’ and ‘MD2’. Group III consisted of nine *Ac(MD2)XTH*, eight *Ac(F153)XTH*, and seven *AtXTH* genes, and was further subdivided into Group IIIA and Group IIIB, as described previously by Baumann et al. [10]. Group IIIA contained only *Ac(F153)XTH24*. The Ancestral Group contained the fewest members group, namely *Ac(F153)XTH1* and *Ac(MD2)XTH1*, whereas Group I/II contained the most members in each cultivar.

### 3.3. Sequence Alignment of XTHs

Multiple sequence alignment showed that *Ac(MD2)XTHs* and *Ac(F153)XTHs* genes shared a highly conserved domain containing the motif ExDxE (Appendix A) [12]. One potential N-linked glycosylation site sharing N(T)-L(K/R/V/T/I)-S(T)-G(N) was located close to catalytic residues in 30 XTHs [49]. In addition, a conserved DWATRGG motif and Cys residues were located in the C-terminal region.

### 3.4. Gene Structure Analysis and the Pattern of the Motif in XTHs 

Highly structural similarity was evident in each phylogenetic group of XTHs. The exon number varied from three to seven in ‘F153’ and ‘MD2’ (Appendix A). Three or four exons were observed in majority of XTHs. Group I/II comprised three or four exons, except that *Ac(MD2)XTH5* contained seven 7 exons. Fourteen of the 17 genes in Group III possessed four exons, whereas *Ac(MD2)XTH19*, *Ac(MD2)XTH20* and *Ac(F153)XTH23* comprised three, three and five exons, respectively. *Ac(MD2)XTH1* with three exons and *Ac(F153)XTH1* with four exons were placed in Group IV. *Ac(MD2)XTH5* with seven exons was longer than all other members (Appendix A), because it possessed four highly conserved domains: two GH16_XET and two C-XET domains (Figure 2). 

Almost all XTHs within the same group shared common motifs (Figure 2). Motif1-2 and motif5 were highly conserved in all XTHs. Motif1-7 and motif 10 were present in Group I/II except that *Ac(MD2)XTH12* lacked motif3 and five genes contained the additional motif8. *Ac(MD2)XTH5* contained 16 motifs in accordance with its structure. *Ac(F153)XTH24* shared motif1-7 in Group IIIA and motifs without rules (6–9 motifs) were presented in Group IIIB. In Group IV, *Ac(MD2)XTH1* possessed the fewest motifs and the motif composition of *Ac(F153)XTH1* was identical to that of Group I/II.

### 3.5. Chromosomal Distribution and Syntenic Analysis of XTH Genes 

Twenty-four *Ac(F153)XTHs* genes were unevenly distributed in 14 of 25 linkage groups (LG) in ‘F153’. LG06 contained the most *Ac(F153)XTHs* genes (Figure 3A). Three linkage groups, consisting of LG03, LG14, and LG15, shared more than two XTH members, whereas only one gene was discovered on each of the remaining chromosomes. All *Ac(MD2)XTH* genes were mapped on different scaffolds in ‘MD2’ (Appendix A).

To analyze duplication events, we detected syntenic blocks using MCScanX in ‘F153’ and ‘MD2’. Thirteen collinear pairs, including 14 *Ac(F153)XTHs* and 12 *Ac(MD2)XTHs* genes, were discovered through synteny analysis (Figure 3B, Appendix A). Almost all pairs represented segmental duplication without tandem duplication and were placed in the same phylogenetic group. The gene pair of *Ac(F153)XTH6-Ac(MD2)XTH8* was placed in Group I/II and *Ac(F153)XTH20-Ac(MD2)XTH23* was placed in Group IIIB, which indicated that XTHs generated multiple segmental duplications have occurred during XTH diversification in pineapple.

### 3.6. Ka/Ks Analysis of XTH Genes 

To assess whether XTH genes had been subject to Darwinian selection, all paralogous XTH pairs were used to calculate Ka/Ks values (Appendix A). Thirteen XTH paralogous with high similarity were detected with the Ka/Ks < 1, which suggested that XTHs had undergone strong purifying selection in ‘F153’ and ‘MD2’. Most paralogous showed a relatively recent duplication time with an average value of about 1.7 Mya, except that *Ac(F153)XTH18-Ac(MD2)XTH18* and *Ac(F153)XTH7-Ac(F153)XTH18* diverged about 388 and 340 Mya, respectively (Table 2).

### 3.7. Differential Expression Profiles of XTHs during Development

Fifteen XTH genes were simultaneously induced in the green leaf tip and white leaf base, which were used to investigate diurnal expression patterns (Figure 4A). The majority of XTHs expressed in the green leaf tip showed transcript levels lower than those detected in the white leaf base or no transcripts were detected. Four genes (27%), comprising *Ac(MD2)XTH6*, *11*, *15*, and *20* exhibited higher transcript levels. Interestingly, several XTHs may show a diurnal expression pattern owing to the contrasting transcript levels detected during day and night in each tissues. 

Fifteen *Ac(MD2)XTHs* genes, which were divided into three diverse categories, were detected during fruit ripening (Figure 4B). Five genes showed higher transcript levels at the Fruit1 and/or Fruit2 stages than that at advanced stages of ripening. *Ac(MD2)XTH6*, *15*, *20*, and *22* showed normal transcript levels or no significant difference among immature stages, but showed a high transcript level in Fruit3 and subsequently a low transcript level during fruit maturation stages. Six XTHs were showed a low transcript levels at the onset of maturity and subsequently were highly expressed. Notably *Ac(MD2)XTH8* and *Ac(MD2)XTH23*, for which expression increased more than 6-fold and 11-fold, respectively, were indicated to have important roles in fruit ripening.

### 3.8. RT-qPCR Analysis of XTH Genes in Root and Leaf 

Quantitative real-time PCR analysis was used to analyze the expression patterns of 11 selected *Ac(MD2)XTH* genes in the root and leaf of ‘MD2’. Eight of the 11 *Ac(MD2)XTH* genes showed differential expression patterns in different tissues, and expression of the remaining three *Ac(MD2)XTH* genes was not detected (Figure 5, Appendix A). *Ac(MD2)XTH15* and *Ac(MD2)XTH18* were detected simultaneously in three tissues. *Ac(MD2)XTH15* showed the highest relative expression level in the green leaf tip, sequentially in the root, and finally in the white leaf base. *Ac(MD2)XTH18* was significantly more highly expressed in the root compared with the other tissues (Figure 5A). Four genes (*Ac(MD2)XTH11, 15, 18,* and *20*) showed higher relative expression levels in the green leaf than those in the white leaf base consistent with the corresponding transcriptome except for *Ac(MD2)XTH18* (Figure 5B, Appendix A). Six genes were observed to show a low relative expression level in the white leaf base, of which the highest was *Ac(MD2)2XTH13* about 0.12 (Appendix A). The relative expression level of four genes ranged from 0.13 to 49.5 in the green leaf tip. In the root, *Ac(MD2)XTH18* showed the highest relative expression level of about 532.5.

## 4. Discussion 

As a vital cell-wall-modifying enzyme, XTH is implicated in the incorporation or hydrolysis of XyG to regulate cell wall remodeling and degradation. In this study, 48 (24/24) non-redundant XTHs were identified in pineapple ‘F153’ and ‘MD2’, respectively. The number of XTHs identified is fewer than that in a previous study because of the redundancy of annotation [15]. The number of XTHs detected in each cultivar is less than that reported in most vascular plant species except *Actinidia deliciosa* (14 *AdXTHs*) and *Malus sieversii* (11 *MsXTHs*) [24]. The difference may be attributed to fewer duplication events of pineapple compared with most other plant species. Gene duplication provides a source for gene functional diversification and contributes to amplification of the number of members of a gene family. For example, an ancient whole-genome duplication (WGD) with massive gene duplication were occurred in *O. sativa* and the genome of *A. thaliana* primarily experienced at least four large-scale duplication events [50,51]. Therefore, the limited number of XTHs in pineapple might reflect that the genome diverged prior to the Poales-specific ρWGD event [2].

The groups retrieved in the phylogenetic tree contained a similar number of XTH genes from each cultivar, which indicated that XTH genes were relatively conserved before the differentiation of cultivars in pineapple. Although differing considerably in MW, PI, and length, the XTHs contained relatively conserved motifs and gene structure in each group, indicating that XTHs of the same group may perform similar functions. In addition, unique traits of XTHs were confirmed in pineapple. For instance, the catalytic sites, D(N)E(L/I/V)DF(Y)EFLG as motif B, were highly conserved in all XTH proteins, especially three absolutely conserved catalytic residues (ExDxE) [12], suggesting that the highly conserved motif could have a similar and conserved function in plant XTHs [13,52]. All proteins were localized to the cell wall, and several proteins were also localized to the cytoplasm. These results indicated that XTH proteins could easily access the substrate and promote catalytic reactions. A signal peptide of about 20–25 nucleotides adjacent to the start codon was present in the majority of XTHs, which suggested that this short sequence of hydrophobic amino acids may be responsible for transmembrane transport of XTH proteins through interaction with subcellular organelle membranes [53]. Motif 9 distinguished Group IIIB from the other groups, and thus might play an important role in the distinctive function of Group IIIB XTHs. Several Cys residues present in the C-XET domain may contribute to the structural stabilization through disulfide bonds [12,54]. Therefore, the conservation of features of the domains, motifs, and gene structure strongly supported the reliability of phylogenetic classification and especially the close evolutionary relationship of two cultivars. 

Group I/II was the biggest group of XTHs, whereas Group IIIA was the smallest group in each cultivar, which was consistent with other species [10]. The XEH activity of Group IIIA evolved as a-gain-of function in ancestral XET [6,10]. However, no XTH from ‘MD2’ was placed in Group IIIA. Archetypal XTH from *Tropaeolum majus*, *Vigna angularis*, and *A. thaliana* in Group IIIA only showed demonstrable XEH activity [14,55,56]. Hence, we speculated that *Ac(F153)XTH24* in Group IIIA might possess XEH activity. 

Twenty-six of the 48 XTHs identified were involved in the segmental duplication events without tandem duplication, which implied that segmental replication events played an important role in XTH gene family expansion and greatly drove the evolution of XTHs in pineapple. Tandem duplication events are undoubtedly crucial for genome expansion [57]. Each gene of the *Ac(F153)XTH7-Ac(F153)XTH8* pair was linked and connected with *Ac(MD2)XTH18*. This result indicated that the three XTHs were highly conserved and experienced translocation or segmental replication in ‘F153’ first, and subsequently evolved in parallel in the two cultivars [58]. The paralogous pairs *Ac(F153)XTH7-Ac(F153)XTH18* and *Ac(F153)XTH18-Ac(MD2)XTH18*, which were indicated to have diverged more than 300 Mya, arose before divergence of the Poales [59,60,61]. All XTHs were indicated to have undergone strong purifying selection suggesting that XTHs have evolved slowly between cultivars [62].

Expression of XTHs varies under exposure to stress and shows tissue, organ, and temporal specificity [4,22,25]. The contrasting expression patterns in the green leaf tip and white leaf base was suggestive of tissue specificity of XTHs in pineapple. Pineapple fixes carbon dioxide nocturnally by activity of CAM-related enzymes and is stored rapidly as malic acid in the vacuole [2,63]. *Ac(MD2)XTH15, Ac(MD2)XTH11, Ac(MD2)XTH6,* and *Ac(MD2)XTH20* showed higher expression levels in photosynthetic tissues and may be putative CAM-related genes that enhance the efficiency of water use by the CAM pathway. Several XTHs participated in fruit ripening, such as in apple and kiwifruit [24]. Together with fruit ripening, continuously fluctuating expression patterns with gradual decline or accumulated increment indicated that XTHs showed polygenic interaction and temporal expression during fruit development stages. In addition, the RT-qPCR results were highly consistent with the transcriptomic data. The results revealed that the expression pattern of *Ac(MD2)XTH15* differed from that of *Ac(MD2)XTH18* in three tissues consistent with placement of the two genes in different phylogenetic groups. An extremely high expression level of *Ac(MD2)XTH18* in the root at noon suggested this gene may be involved in root growth during photosynthesis. These results indicated that several XTHs are involved in photosynthesis with tissue specificity as putative CAM-related enzymes. 

## Figures and Tables

**Figure 1 genes-10-00537-f001:**
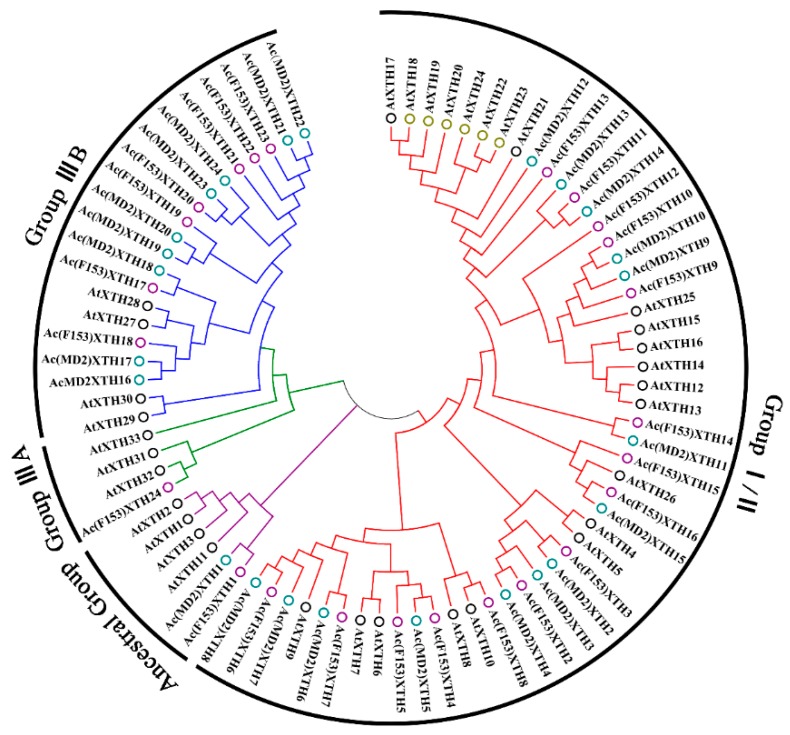
The classification of XTH genes in ‘F153’, ‘MD2’ and *A. thaliana*. Four colorful braches with red, green, blue, and purple were showed Group I/II, IIIA, IIIB and Ancestral Group, respectively. Circles of different color represented the kinds of species (‘MD2’ with blue circles, ‘F153’ with purple circles and *A. thaliana* with black circles).

**Figure 2 genes-10-00537-f002:**
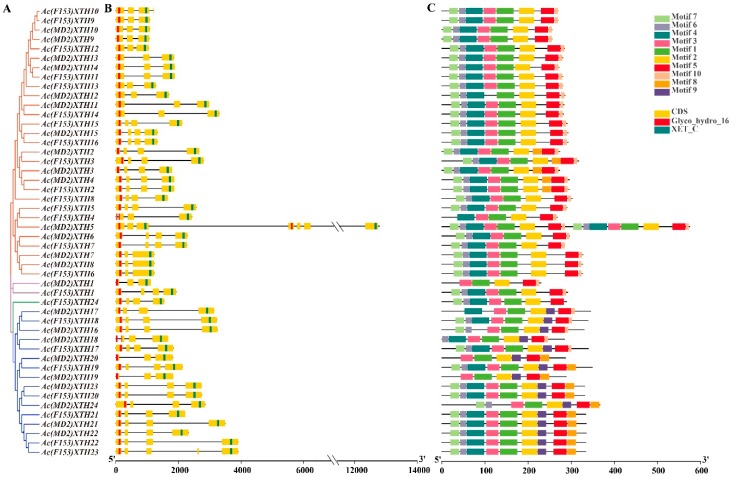
Phylogenetic relationships, gene structure, and motifs distribution of XTHs. (**A**) The phylogenetic tree was highlighted by different colors with Group I/II, IIIA, IIIB and Ancestral Group; (**B**) Gene structure and conserved domains. Yellow boxes and black lines represented exons and introns, respectively. The conserved domains were highlighted by red and blue strips; (**C**) The motif distribution. Motif1-10 in different colorful boxes.

**Figure 3 genes-10-00537-f003:**
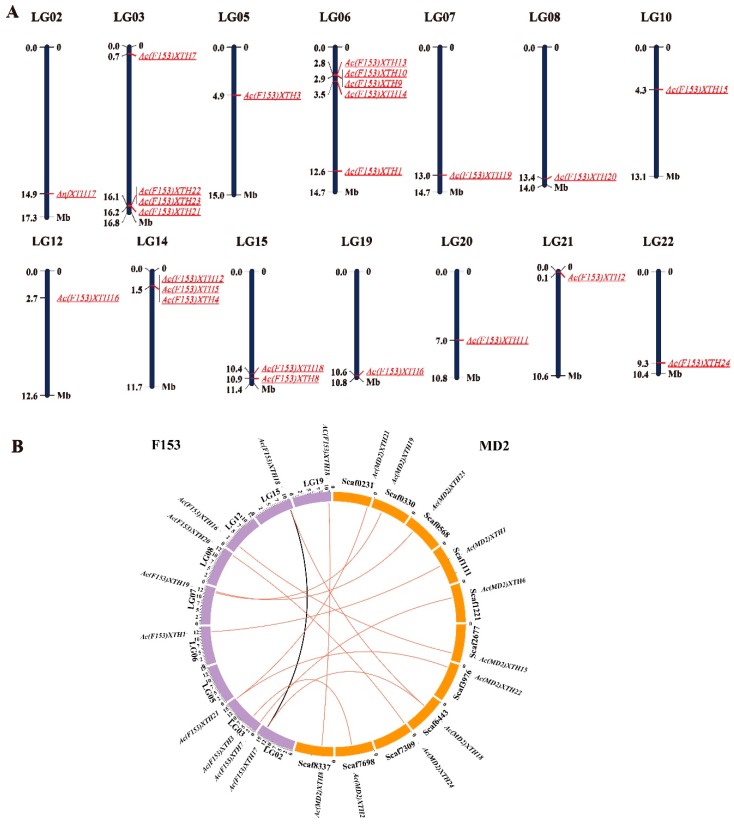
The chromosome distribution and synteny analysis of XTHs in pineapple. (**A**) There were 24 *Ac(F153)XTHs* on each chromosome in ‘F153’. Each pillar represented a chromosome and the scale bar was set in mega base (Mb). The gene names were shown on each chromosome with red; (**B**) Syntenic relationships among ‘F153’ and ‘MD2’. The links represented different gene replications across cultivars or chromosomes. ‘F153’ marked by the purple arcs and ‘MD2’ marked by the orange arcs.

**Figure 4 genes-10-00537-f004:**
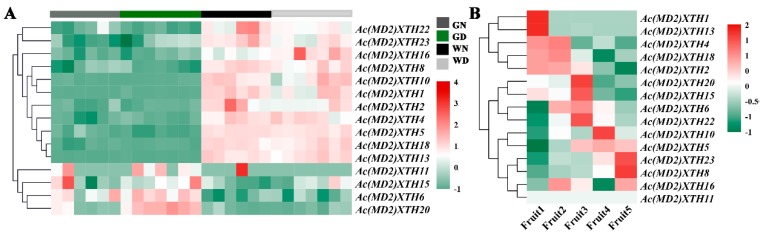
The expression patterns during development in ‘MD2’. (**A**) The expression in the green leaf tip and white leaf base during thirteen time points over a 24-h period. GN was displayed green leaf tip during nighttime, GD was displayed green leaf tip during daytime, WN was displayed white leaf base during nighttime, WD was displayed white leaf base during daytime; (**B**) The expression levels during fruit development. Fruit1, Fruit 2 and Fruit 3 were indicated immature stages, Fruit 4 and Fruit5 displayed maturity stages.

**Figure 5 genes-10-00537-f005:**
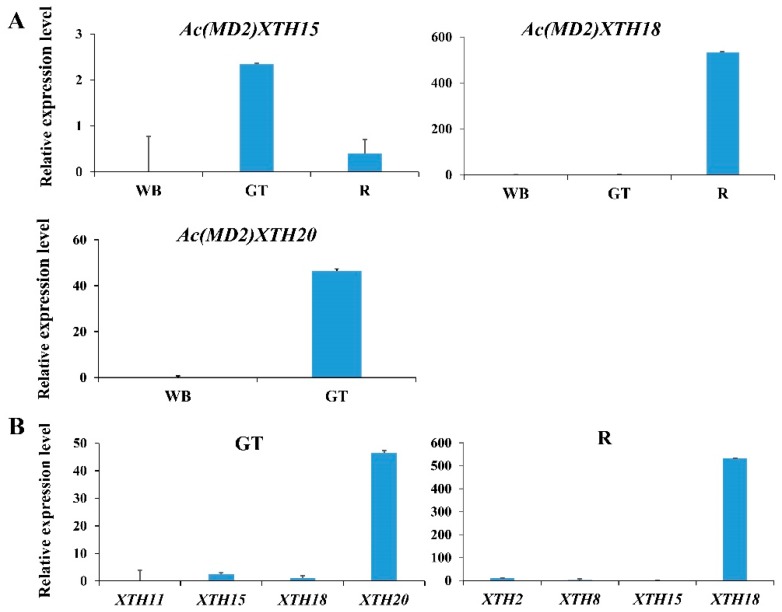
The relative expression level in different tissues of ‘MD2’ by RT-qPCR. WB, GT, and R represented white leaf base, the green leaf tip, and root, respectively. (**A**) The expression levels in the green leaf tip, white leaf base, and (or) root; (**B**) Expression levels of several *Ac(MD2)XTH* genes in the green leaf tip and root.

**Table 1 genes-10-00537-t001:** The physicochemical properties of XTHs from ‘F153’ and ‘MD2’.

Name	Transcript ID	Length	MW (kDa)	PI	SP	Catalytic Site	Subcellular Localization
*Ac(F153)XTH1*	XP_020091206.1	292	32.78	5.35	20	DELDFEFLG	Cell wall
*Ac(F153)XTH2*	XP_020111283.1	296	34.17	5.98	20	DEIDFEFLG	Cell wall Cytoplasm
*Ac(F153)XTH3*	XP_020087864.1	318	37.29	7.59	−	DEIDFEFLG	Cell wall Cytoplasm
*Ac(F153)XTH4*	XP_020102739.1	268	30.71	5.96	−	DELDFEFLG	Cell wall
*Ac(F153)XTH5*	XP_020102738.1	291	33.33	6.06	24	DELDFEFLG	Cell wall
*Ac(F153)XTH6*	XP_020109096.1	327	35.98	5.07	24	NEFDFEFLG	Cell wall
*Ac(F153)XTH7*	XP_020084286.1	285	31.65	5.53	27	DEVDFEFLG	Cell wall
*Ac(F153)XTH8*	XP_020104828.1	303	35.62	8.81	−	DEIDFEFLG	Cell wall
*Ac(F153)XTH9*	XP_020089756.1	270	30.56	6.1	20	DEIDFEFLG	Cell wall Cytoplasm
*Ac(F153)XTH10*	XP_020090359.1	270	30.63	6.43	20	DEIDFEFLG	Cell wall Cytoplasm
*Ac(F153)XTH11*	XP_020110218.1	281	31.54	4.69	21	DEIDFEFLG	Cell wall Cytoplasm
*Ac(F153)XTH12*	XP_020102740.1	285	31.58	4.75	24	DEIDFEFLG	Cell wall Cytoplasm
*Ac(F153)XTH13*	XP_020091231.1	281	31.46	5.71	23	DEIDFEFLG	Cell wall Cytoplasm
*Ac(F153)XTH14*	XP_020090869.1	282	32.15	5.41	25	DEVDFEFLG	Cell wall
*Ac(F153)XTH15*	XP_020097886.1	291	33.33	4.8	26	DEIDYEFLG	Cell wall
*Ac(F153)XTH16*	XP_020100605.1	293	32.89	9.04	20	NEVDFEFLG	Cell wall
*Ac(F153)XTH17*	XP_020106929.1	340	38.92	5.95	28	DELDFEFLG	Cell wall
*Ac(F153)XTH18*	XP_020104936.1	339	38.68	6.56	26	DELDFEFLG	Cell wall
*Ac(F153)XTH19*	XP_020092864.1	349	39.41	8.69	23	DELDFEFLG	Cell wall
*Ac(F153)XTH20*	XP_020094226.1	331	37.68	5.79	20	DELDFEFLG	Cell wall
*Ac(F153)XTH21*	XP_020085280.1	334	37.63	5.93	19	DELDFEFLG	Cell wall
*Ac(F153)XTH22*	XP_020085278.1	334	37.65	5.93	19	DELDFEFLG	Cell wall
*Ac(F153)XTH23*	XP_020085279.1	334	37.65	5.93	19	DELDFEFLG	Cell wall
*Ac(F153)XTH24*	XP_020112380.1	290	33.23	9.53	18	DEVDIEFLG	Cell wall
*Ac(MD2)XTH1*	OAY79161.1	230	25.99	5.36	−	DELDFEFLG	Cell wall
*Ac(MD2)XTH2*	OAY64709.1	274	32.15	7.63	−	DEIDFEFLG	Cell wall Cytoplasm
*Ac(MD2)XTH3*	OAY72845.1	274	31.83	5.98	−	DEIDFEFLG	Cell wall Cytoplasm
*Ac(MD2)XTH4*	OAY65283.1	296	34.17	5.98	20	DEIDFEFLG	Cell wall Cytoplasm
*Ac(MD2)XTH5*	OAY76125.1	575	64.58	5.26	24	DELDFEFLG	Cell wall Cytoplasm
*Ac(MD2)XTH6*	OAY78767.1	296	32.87	5.7	38	DEVDFEFLG	Cell wall
*Ac(MD2)XTH7*	OAY70160.1	328	36.16	5	24	NEFDFEFLG	Cell wall
*Ac(MD2)XTH8*	OAY63484.1	327	35.98	5.07	24	NEFDFEFLG	Cell wall
*Ac(MD2)XTH9*	OAY62696.1	256	29.06	6.1	−	DEIDFEFLG	Cell wall Cytoplasm
*Ac(MD2)XTH10*	OAY67076.1	256	29.06	6.1	−	DEIDFEFLG	Cell wall Cytoplasm
*Ac(MD2)XTH11*	OAY62698.1	284	32.34	5.41	27	DEVDFEFLG	Cell wall
*Ac(MD2)XTH12*	OAY79036.1	286	32.28	5.97	25	DEIDFEFLG	Cell wall
*Ac(MD2)XTH13*	OAY76653.1	281	31.57	4.69	21	DEIDFEFLG	Cell wall Cytoplasm
*Ac(MD2)XTH14*	OAY71418.1	272	30.48	4.68	21	DEIDFEFLG	Cell wall
*Ac(MD2)XTH15*	OAY73488.1	293	32.89	9.04	20	NEVDFEFLG	Cell wall
*Ac(MD2)XTH16*	OAY70295.1	330	37.64	6.6	26	DELDFEFLG	Cell wall
*Ac(MD2)XTH17*	OAY81259.1	345	38.76	5.63	−	DELDFEFLG	Cell wall
*Ac(MD2)XTH18*	OAY66122.1	285	32.60	6.22	−	DELDFEFLG	Cell wall
*Ac(MD2)XTH19*	OAY83621.1	289	32.55	8.57	−	DELDFEFLG	Cell wall
*Ac(MD2)XTH20*	OAY70631.1	287	32.53	8.26	−	DELDFEFLG	Cell wall
*Ac(MD2)XTH21*	OAY84325.1	335	37.79	6.17	19	DELDFEFLG	Cell wall
*Ac(MD2)XTH22*	OAY70279.1	335	37.80	6.17	19	DELDFEFLG	Cell wall
*Ac(MD2)XTH23*	OAY81925.1	331	37.68	5.95	20	DELDFEFLG	Cell wall
*Ac(MD2)XTH24*	OAY65080.1	367	41.61	8.54	−	DELDFEFLG	Cell wall

MW: molecular weight; PI: isoelectric point; SP: Signal Peptide.

**Table 2 genes-10-00537-t002:** Ka/Ks analysis and estimated divergence time of XTHs.

Collinear XTH Pairs	Ka	Ks	Ka/Ks	*p*-Value (Fisher)	Duplication Time (Mya)
*Ac(F153)XTH7-Ac(MD2)XTH6*	0.003077	0.03004	0.102442	0.003098	2.46
*Ac(F153)XTH21-Ac(MD2)XTH22*	0.003969	0.024998	0.158756	0.008438	2.05
*Ac(F153)XTH21-Ac(MD2)XTH21*	0.006628	0.029215	0.226877	0.011896	2.39
*Ac(F153)XTH3-Ac(MD2)XTH2*	0.00155	0.011451	0.135322	0.107858	0.94
*Ac(F153)XTH1-Ac(MD2)XTH1*	0.008336	0.010261	0.812454	0.377377	0.84
*Ac(F153)XTH19-Ac(MD2)XTH19*	0.007705	0.019414	0.396873	0.150722	1.59
*Ac(F153)XTH7-Ac(MD2)XTH12*	0.062047	0.071925	0.862666	0.630505	5.90
*Ac(F153)XTH20-Ac(MD2)XTH24*	0.001324	0.008482	0.15611	0.130065	0.70
*Ac(F153)XTH16-Ac(MD2)XTH15*	NA	0.005103	0	NA	0.41
*Ac(F153)XTH18-Ac(MD2)XTH18*	0.265729	4.64275	0.057235	1.19E^−40^	380.55
*Ac(F153)XTH17-Ac(MD2)XTH18*	NA	NA	NA	NA	NA
*Ac(F153)XTH7-Ac(F153)XTH18*	0.311919	4.15239	0.075118	1.28E^−41^	340.36
*Ac(F153)XTH6-Ac(MD2)XTH18*	NA	NA	NA	NA	NA

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
