# Peer review of "Genome-Wide Identification and Characterization of Xyloglucan Endotransglycosylase/Hydrolase in Ananas comosus during Development"

_genes, 2019, doi:10.3390/genes10070537_

Round 1
Reviewer 1 Report
This manuscript investigated the phylogenetic relationships, diversity, and predicted structural complexity among 48 XTH genes in two Ananas sp, A main objective of the manuscript was to consolidate genomic and database (e.g. transcriptomics) information to provide a single comprehensive picture of XTHs in Ananas comosus. The manuscript utilized bioinformatics and ‘genomic surveys’ to conduct XTH phylo-comparison and analysis. In the past, I have often found bioinformatics articles like the manuscript under review here to be quite useful however, I have several concerns with the current manuscript and these are outlined below.
The authors used an open accessed database to know the expression pattern of XTHs. While knowing their expression patterns is certainly of value, simply presenting mined data from databases, without some sort of corroboration is dubious and of limited use. The experimental verification is absolutely necessary. Just organizing expression data procured from a primary database search into heatmaps have no-novelty and no-worth. Hence, I suggest the authors do experimental verification via qRT-PCR, at least for few significantly expressed XTHs.
Authors can also provide some important analyses like the rate of non-synonymus substitution (Ka), the rate of synonymous substitutions (Ks) and Ka/Ks to develop a better understanding of the evolution of the family among different genomes (including Arabidopsis, Rice, Nicotiana, and other fruit species), it may be helpful to correlate gene family analysis with different duplication modes and can be determined for a more integrated view. For example, you could see and cite a recent paper: Comprehensive genome-wide survey, genomic constitution and expression profiling of the NAC transcription factor family in foxtail millet (Setaria italica L.) by Puranik et al. PLoS ONE 8(5): e645943.
Often, I was able to guess the intent but scientific writing should minimize interpretability not confound it. Correction of these grammatical issues is addressable but certainly, in the current form, the English in the manuscript does not achieve a standard of acceptable quality. As a whole, the manuscript is poorly organized, for example
MS is strongly lacking in the results and discussion of their hypothesis.
Basic information on the cultivars F153 and MD2 e.g. origin and other morphological features are missing.
Table-1 is missing in the MS,
Arabidopsis should be Arabidopsis thaliana at its first appearance and later A. thaliana, elsewhere in the text
………“functional characterization on genetics” ??? I assume genes need functional characterization.
The physicochemical properties of XTH in two cultivar pineapple…should be “…in two pineapple cultivars”
Author Response
Dear Editor,
Thank you very much for your letter and the comments from the referees about our paper submitted to Genes (Manuscript ID: genes-503607).
We have learned much from the reviewers’ comments, which are fair, encouraging and constructive. After carefully studying the comments and your advice, we have made corresponding changes. Our response of the comments is enclosed at the end of this letter. We also add some experiments in this manuscript.
We deeply appreciate your consideration of our manuscript, and we look forward to receiving comments from the reviewers again.
If you have any queries, please don’t hesitate to contact me at the address below.
Best wishes,
Sincerely yours,
Long Yang
Lyang@sdau.edu.cn

Reviewer 2 Report
This manuscript by Li et al. presents a systematic survey of XTH genes in pineapple and some rudimentary expression analysis during development. Overall, the work makes a minor contribution to our understanding of XTH function, yet the description of the sequence structure of XTHs in this organism may be of some use to researchers interested in these cell wall enzymes.
The manuscript is logical and, overall, clearly presented. The following comments must be addressed to improve the manuscript further:
The manuscript suffers significantly from poor grammar throughout. It must be revised by a professional editing service or a native English speaker.
1. Line 16 and onward: The abbreviated names for the XTHs across cultivars is confusing (Anf vs. Anm). Use instead "Ac(F153)XTH" and "Ac(MD2)XTH" throughout. (See Henrissat et al. FEBS Lett. 1998 Mar 27;425(2):352-4 for the usual practice for carbohydrate-active enzymes).
2. Lines 15-17 and onward: The authors do not seem to be aware of the study by Behar et al. [Plant J. 2018 Sep;95(6):1114-1128. doi: 10.1111/tpj.14004], which provides a survey of XTH in Ananas comosus (see especially Fig. 6). The number of XTH found here should be compared with this earlier work.
3. Related to the previous comment: Please also identify any EG16 members in the F153 and MD2 genomes, including indicating their chromosome positions and homology (Fig. 3) and gene structure (Fig. 2). Plant EG16 are also GH16 members, and are closely related to the XTH.
4. Line 83: Cite Behar et al. Plant J. 2018 Sep;95(6):1114-1128. doi: 10.1111/tpj.14004 after "....publicly available datasets,"
5. Lines 86-87: With reference to comment 3 above, this statement does not seem to be true. However, I note that the authors of the present manuscript do a more in depth analysis of pineapple XTH phylogeny, gene structure, and chromosome location, which brings additional value. Please revise.
6. Lines 122-127: Overall, the NJ method is one of the least robust in molecular phylogeny. The analysis should be repeated using Maximum Likelihood analysis (PhyML, RAxML, or similar). Also, please supply the curated input alignment file in FASTA format, to allow others to verify and build-upon this work.
7. Lines 133-138 and Fig. 3: Please number all XTH in order according to their positions on the chromosomes (start with XTH1 on LG02 in Fig. 3). This would make for a far more systematic and logic numbering scheme; following the Arabidopsis numbering (lines 153-154) does not make sense, as Arabidopsis is distantly related and has several more XTH. In Fig. 3, please also show the chromosome loci for the MD2 cultivar. Including a table of orthologous sequences between the cultivars would be helpful. (I do not see Table 1 in the version provided, Line 154, 156). Please also include more rich discussion of orthology, supported by Figures 1 and 3.
8. Line 152 and throughout. Please revise the active-site motif to "ExDxE", since not all of the residues shown are fully conserved, and the shorter version contains the key catalytic residues. See and cite here: Johansson et al. Plant Cell. 2004 Apr;16(4):874-86; and Baumann et al. Plant Cell. 2007 Jun;19(6):1947-63
9. Lines 152-155: State how the number of XTH found compare to Behar et al. Plant J. 2018 Sep;95(6):1114-1128. doi: 10.1111/tpj.14004 and describe any differences/similarities. If EG16 sequences are found (see comment 3 above), please indicate this here.
10. Figure 1: As indicated above, please supply the curated alignment file used as input for the phylogenic analysis as a supplementary file in FASTA format.
11. Line 166: "signal peptides"?
12. Line 160-166 (also lines 307-316): If EG16 sequences are found (see comment 3 above), please describe their properties. Please also include a supplemental alignment file (FASTA) conrtaining any pineapple EG16s and other representative EG16s (see also Eklof et al J Biol Chem. 2013 May 31;288(22):15786-99. doi: 10.1074/jbc.M113.462887 and McGregor et al. Plant J. 2017 Feb;89(4):651-670. doi: 10.1111/tpj.13421
13. Lines 168-170 and Fig. 2 (also lines 292-295): Please clearly describe how these groups compare to those established by Yokoyama, Rose, and Nishitani (ref. 21). Given their proposal to merge Groups I and II, is there any reason to separate them here? Any new group delineations must be supported by strong bootstrap values from ML phylogenetic analysis.
14. Lines 173-174 (also lines 300-302): Describe the significance of Group IIIA membership with regard to XEH activity. See see Baumann et al. Plant Cell. 2007 Jun;19(6):1947-63 ; Kaewthai et al. Plant Physiol. 2013 Jan;161(1):440-54 ; Eklof et al. Plant Physiol. 2010 Jun;153(2):456-66. doi: 10.1104/pp.110.156844
15. Lines 178-182: In addition to ref. 45, more modern are needed regarding enzyme structure, mechanism, and glycosylation: Johansson et al. Plant Cell. 2004 Apr;16(4):874-86; and Baumann et al. Plant Cell. 2007 Jun;19(6):1947-63 ; Saura-Vallas et al. Biochem J. 2006 Apr 1;395(1):99-106 ; Kallas et al. Biochem J. 2005 Aug 15;390(Pt 1):105-13
16. Lines 208-219 (also lines 321-322): This section would fit better under Section 3.3. Also, the value of defining 10 motifs is unclear. Highlight which ones correspond to the GH16 domain, the catalytic active-site, the XEH/Group III-specific loop, and the XET_C domain, with reference to Johansson et al. Plant Cell. 2004 Apr;16(4):874-86; and Baumann et al. Plant Cell. 2007 Jun;19(6):1947-63; McGregor et al. Plant J. 2017 Feb;89(4):651-670. doi: 10.1111/tpj.13421 ; https://pfam.xfam.org/family/PF06955
17. Essentially all of the figures are too small/too poor resolution to be readable.
18. Lines 43-44: Cite here: Eklof et al. Plant Physiol. 2010 Jun;153(2):456-66. doi: 10.1104/pp.110.156844
End of comments.
Author Response

(The authors gave the same response as above.)

Round 2
Reviewer 1 Report
Authors have done substantial improvements in the MS, hence it could be excepted for publication.
Author Response
Dear Editor,
Thank you very much for your letter and the comments from the referees about our paper submitted to Genes (Manuscript ID: genes-503607).
We are all extend our sincere appreciation for the reviewer's positive comments. Thank you for taking time to review the modified manuscript. We are pleased to note the favorable comments of reviewers in your comments of all of research. Thanks for the reviewer’s attention to our amended manuscript and we would like to appreciate your approbatory evaluation again. We appreciate for reviewers’ warm work earnestly, special thanks for your approval.
If you have any queries, please don’t hesitate to contact me at the address below.
Best wishes,
Sincerely yours,
Long Yang
Lyang@sdau.edu.cn
Reviewer 2 Report
In this revision, the authors have made a considered effort to address my comments. Some of these have been addressed satisfactorily, while others require further revision.
1. In their response, the authors state that the manuscript has been professionally edited. Given the very many and significant grammatical errors, this is unlikely, or the authors have received a very poor quality service from whatever company they used. Regrettably, in many places the grammar is so poor that it is not clear what the authors are trying to say. For example (Abstract, line 29-30): "This study will be responsible for understanding molecule basis and functional characterization in A.c." There are many, many such examples. This is simply scientifically incorrect: "...but 11 other chromosomes weren't found in any XTHs" (Line 228-299).
* Before it can be accepted for publication, the manuscript must be revised (again) by a native English speaker or a reputable professional editing service (e.g. https://www.sees-editing.co.uk/ )
2. With reference to my previous comments 5 and 14, regarding the existence of an EG16 member in the pineapple genome:
Using tBLASTn on NCBI I searched through the assemblies used in the study (accession numbers: GCA_001540865.1 and GCA_001661175.1) with the AcEG16 sequence from the Behar et al. publication.
>AcEG16
MAESPSPLPIPNEHHHGTALLREIAVDYCPEACDHSPAASEIRVVFDHRGGARWRSRSRFRFGTFGALIRGPGGDTSGLNFNLYLSSLEGDKSQDEIDFEFLGNDPSAVQTNFYTAGLGRRERIHPLGFHAADGFHEYLIKWAPHLIEWLVDGAVLRREERSAAHQPWPLKPMFLYASVWDASYIDDGRWTGSYVGSDAPYVCLYRDVRVPIDNAVAEEGDAAAEAAAAAAAEAEKQNVS
In both genomes I got the exact sequence back (100% query coverage and 100% sequence identity).
AcMD2: Ananas comosus cultivar MD2 ACMD2_Scaffold_1399, whole genome shotgun sequence. Sequence ID: LSRQ01001399.1, Range: 9655821 to 9656540
AcF153: Ananas comosus cultivar F153 linkage group 8, ASM154086v1, whole genome shotgun sequence. Sequence ID: NC_033628.1 Range: 9655821 to 9656540.
NCBI Conserved Domain analysis validates that this sequence has a GH16 domain (pfam00722) and GH16_XET domain but no XET_C domain.
Therefore, the authors' response to my previous comment 14 (sub-response 2) that their is no EG16 homolog ("...owing to absence of EG16") is inaccurate.
* Please resolve this situation by adding a sentence at the end of section 3.1 stating, for example: "In addition, EG16 homologs, which are related to XTH in GH16 but lack the XET_C extension, were found in cultivars MD2 and F153 (Sequence ID: LSRQ01001399.1, Range: 9655821 to 9656540 and Sequence ID: NC_033628.1 Range: 9655821 to 9656540, respectively) [Ref. Behar et al. AND McGregor et al. Plant J. 2017 Feb;89(4):651-670. doi: 10.1111/tpj.13421]"
* Please also indicate in the Abstract that EG16 homologs were found in addition to XTH from GH16. See, for example, https://www.mdpi.com/1420-3049/24/10/1935
Indeed, there are many parallels between the recent publication of Fu et al. [Molecules2019,24(10),
1935] and the present manuscript, such that the authors may wish to align their analysis to make it more comparable. It would also seem to be relevant to cite this work.
3. Line 54: In addition to ref. 12, cite also ref. 10 here.
4. Line 60: After the phrase "An increasing number of XTHs have been identified using publicly available datasets," cite Behar et al.
5. Lines 84-85: My previous comment 7 was not addressed. Revise to "Systematic identification and characterization of XTHs in pineapple has received only limited attention to date [Ref. Behar et al.]"
6. Line 171: After the sentence ending with "...ExDxE", cite current ref. 51 (Johansson et al.) here.
7. Line 188: Revise to "...further subdivided into group III A and group III B, as described by Baumann et al. [ref. 10]"
8. Lines 198-202: The authors' seem to have misunderstood my previous comment - I was simply asking for improved citations to support this section. In this revised version, please:
Change the catalytic motif to "ExDxE" as elsewhere in the text, and cite current ref. 51 (Johansson et al.) here.
Cite Kallas et al. (2015) Biochem J. 390, 105-113 on line 201 to support statements about the glycosylation site.
9. Line 313: Should this be "...E(L/I/V)..." instead of "...EL(I/V)..."?
11. Line 330-332: Revise to "Archetypal XTH from Tropaeolum majus, Vigna angularis, and Arabidopsis thaliana in Group IIIA only showed demonstrable XEH activity [52,53].", additionally including citation of Kaewthai et al. Plant Physiol. 2013 Jan;161(1):440-54. doi: 10.1104/pp.112.207308
12. Line 286: The authors use AnmXTH by mistake instead of Ac(MD2)XTH.
13. Figure 2 is still unreadable due to low image quality
14. Figure 5 does not make sense. The same data is shown twice, for example expression of Ac(MD2)XTH15 in green leaf base (GT) in A and B.
End of comments.
Author Response
Dear Editor,
Thank you very much for your letter and the comments from the referees about our paper submitted to Genes (Manuscript ID: genes-503607).
We have learned much from the reviewers’ comments, which are fair, encouraging and constructive. After carefully studying the comments and your advice, we have made corresponding changes. Our response of the comments is enclosed at the end of this letter.
We deeply appreciate your consideration of revised manuscript, and we look forward to receiving comments from the reviewers again.
If you have any queries, please don’t hesitate to contact me at the address below.
Best wishes,
Sincerely yours,
Long Yang
Lyang@sdau.edu.cn
